# Correlative microscopy approach for biology using X-ray holography, X-ray scanning diffraction and STED microscopy

M. Bernhardt[1], J.-D. Nicolas[1], M. Osterhoff [1], H. Mittelstädt[2], M. Reuss[2], B. Harke[2], A. Wittmeier[1], M. Sprung[3], S. Köster [1] & T. Salditt [1]

We present a correlative microscopy approach for biology based on holographic X-ray imaging, X-ray scanning diffraction, and stimulated emission depletion (STED) microscopy. All modalities are combined into the same synchrotron endstation. In this way, labeled and unlabeled structures in cells are visualized in a complementary manner. We map out the fluorescently labeled actin cytoskeleton in heart tissue cells and superimpose the data with phase maps from X-ray holography. Furthermore, an array of local far-field diffraction patterns is recorded in the regime of small-angle X-ray scattering (scanning SAXS), which can be interpreted in terms of biomolecular shape and spatial correlations of all contributing scattering constituents. We find that principal directions of anisotropic diffraction patterns coincide to a certain degree with the actin fiber directions and that actin stands out in the phase maps from holographic recordings. In situ STED recordings are proposed to formulate models for diffraction data based on co-localization constraints.

[1] Institut für Röntgenphysik, Universität Göttingen, Friedrich-Hund-Platz 1, D-37077 Göttingen, Germany. [2] Abberior Instruments, Hans-Adolf-Krebs-Weg 1, D-37077 Göttingen, Germany. [3] Deutsches Elektronen-Synchrotron (DESY), Notkestraße 47c, D-22607 Hamburg, Germany. Correspondence and requests for materials should be addressed to T.S. (email: tsaldit@gwdg.de)

Nanoscale imaging links cellular functions to the underlying structure, from organelle down to the molecular level. The cytoskeleton is a perfect example: the long-term goal of quantitative modeling, i.e., predicting forces and mechanical properties, requires structural characterization of protein filaments, bundles, and networks. To this end, super-resolution fluorescence microscopy methods, and in particular stimulated emission depletion (STED) microscopy[1–3], have become enabling tools for cytoskeleton protein networks[4,5] as for bioimaging in general[6]. They have broken the Abbé resolution limit of about 200 nm for visible light, based on the switching between emissive (bright) and non-emissive (dark) states of the fluorophore, either by controlled sequential switching (STED) or in a stochastic manner, as in stochastic optical reconstruction microscopy (STORM)[7] and photoactivation localization microscopy (PALM)[8]. However, a complete structural characterization cannot be based on fluorescent markers alone, since this reduces the specimen down to one or a few labeled constituents. Complementary contrast mechanisms are required in order to cover the entirety of cellular structures, notably the mass or electron density distribution. One would like to probe unlabeled biomolecular assemblies in the cell in similar ways as it is possible for in vitro model systems, for example by small-angle X-ray scattering (SAXS) in solution. In this way, molecular crowding, short range order or packing effects such as in the problem of DNA compactification could be addressed quantitatively in terms of correlation functions and corresponding structure factors.

Today, biological cells are also amenable to X-ray analysis, given the recent progress in X-ray focusing[9–14], analysis of diffraction data[15–17], as well as reconstruction algorithms for coherent imaging[13,18–21]. Different X-ray imaging modalities are well established: X-ray microscopy with Fresnel zone plate lenses[22,23], scanning X-ray fluorescence microscopy[24], scanning SAXS[25–32], coherent diffractive imaging (CDI)[33], ptychography[25,34,35], as well as X-ray holography[36–38]. Correlative imaging approaches have become available more recently, for example by combining X-ray fluorescence microscopy with ptychography[39,40], or with coherent diffractive imaging (CDI)[33]. These achievements, combined with versatile state-of-the-art synchrotron endstations[41–45], have made it possible to perform studies on biological cells in different preparation states, comprising freeze-dried, cryogenically vitrified, chemically fixed and living cells[28,46].

X-ray techniques can in particular contribute by providing a quantitative contrast mechanism based on the interaction with the native electron-density. However, the signals are often challenging to interpret without any model, since the signal is originating from the entire biomolecular ensemble. Contrarily, fluorescent markers can inform about specific molecular components, which in turn can help to formulate these models, as well as constrain parameters in X-ray data analysis. X-ray techniques therefore benefit tremendously from comparative and correlative imaging with the more established fluorescence techniques. Conversely, the X-ray results, in particular the projected electron density maps, help to understand in which background structure the labeled molecules are embedded, as well as indicate whether a specific structure is overly or too sparsely labeled. More generally, several previous studies have shown that important problems of biological cells and tissues can only be solved by a combination of different imaging methods[47–52].

In this work, we present in situ STED and X-ray microscopy recorded at the same synchrotron endstation. Until now, super-resolution optical-microscopy and X-ray microscopy have not been carried out in a correlative manner. For the present work, a specifically designed STED microscope has been installed on the optical table of a nano-focus coherent X-ray imaging endstation, which we have optimized for scanning SAXS, as well as in-line X-ray holography. This configuration allows for the recording of X-ray holograms, diffraction data, and STED images to be gathered from the same cell, and in the same sample environment or mounting. We first introduce the concept of all recording schemes and then show how the different contrast mechanisms inform each other for the example of neonatal rat cardiac tissue cells with a labeled actin cytoskeleton.

## Results

**Instrumentation and implementation.** The experiment was performed using the Göttingen Instrument for Nano-Imaging with X-Rays[45], GINIX, at the P10 undulator beamline of the PETRA III storage ring at the Deutsches Elektronen-Synchrotron (DESY). The undulator beam was monochromatized by a Si(111) channel-cut monochromator to $E_{\text{ph.}} = 13.8$ keV and focused by a set of Kirkpatrick–Baez (KB) mirrors to a focal spot size to a full width at half maximum value of about $\text{FWHM}_{y,z} = 300 \times 300$ nm$^2$ in horizontal ($y$) and vertical ($z$) direction. The photon flux was $I_0 \approx 1.1 \cdot 10^{11}$ ph/s. The three imaging modalities used here are X-ray holography, scanning SAXS (with a nano-focused beam) and STED microscopy, all integrated in a single instrument on the same optical table. For the data shown here, they have been carried out in the following sequence: STED microscopy, X-ray holography, and scanning SAXS.

Figure 1 illustrates the setup with all three recording schemes. For X-ray holography (Fig. 1a (top), the KB-prefocused beam is coupled into a lithographic X-ray waveguide channel[53,54], which is placed in the focal plane for purposes of spatial and coherence filtering[55]. The sample is mounted on a fully motorized sample stage. For recordings, the sample is moved to a defocus plane and is illuminated by the highly coherent exit wave field emanating from the waveguide. Small phase shifts of the X-ray beam caused by the interaction with the object (see top box, inset) lead to a modulated exit wavefront, which thus contains the structural information of the object. Phase information is made visible by free space propagation of the beam behind the object and self-interference of the divergent wave field. Hence, the (near-field) interference pattern is a magnified in-line hologram encoding the local phase shift, or equivalently the projected electron density distribution of the object. Accordingly, all molecular constituents, including unlabeled, as well as labeled moieties, contribute to the phase shifts. The phase and amplitude of the exit wave can then be reconstructed from the recorded holograms by phase retrieval, either based on iterative projection algorithms or in some cases direct Fourier-filtering. As a full field technique, X-ray holography does not require any lateral scanning of the sample. In the present case, holograms are recorded by a scintillator-based fiber-coupled sCMOS detector with a pixel size of $p = 6.5$ μm (2048 × 2048 pixels, Photonic Science, UK), placed approximately 5 m behind the sample. By varying the sample position and thus the focus-to-sample distance $z_1$, typically by a few millimeters to centimeters, the geometrical magnification $M = \frac{z_1 + z_2}{z_1}$ defined by $z_1$ and the sample-detector distance $z_2$ of the setup, the effective pixel size $p_{\text{eff}} = \frac{p}{M}$, as well as the field of view, can be adjusted. The final result is a reconstructed local phase or projected electron density map of the object as detailed further below. Note that for a biological cell at the given photon energy, absorption can be neglected, and the cell can essentially be regarded as a pure phase object.

Scanning SAXS (Fig. 1a (center) probes molecular structures by diffraction. To this end, the waveguide is removed and the sample is translated further upstream into the focal plane of the KB-mirrors. During a recording the sample is successively translated through the focus, see center box (inset), which then leads to an

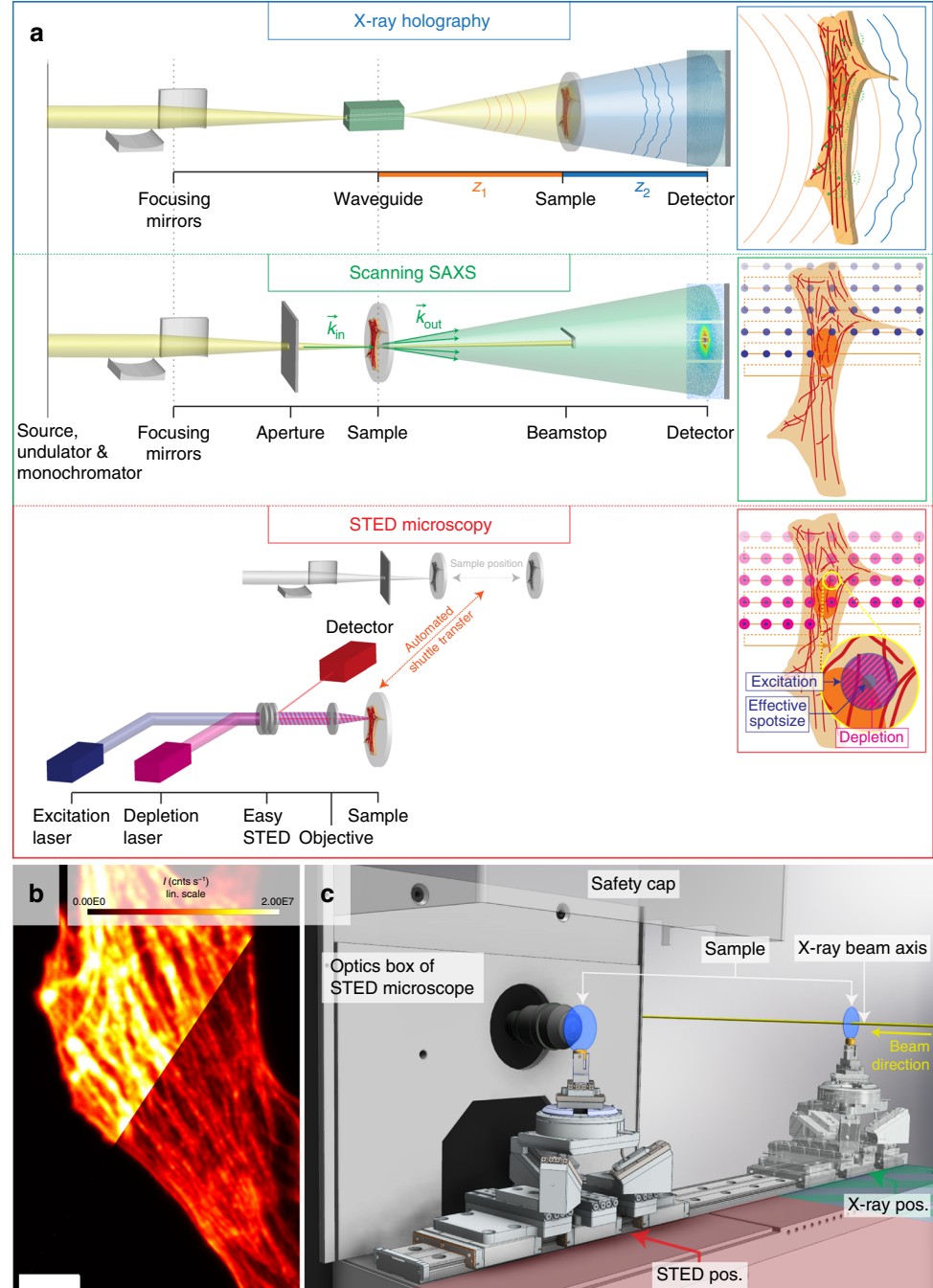

**Fig. 1** Recording schemes and instrumental realization at the beamline. **a** Schematic of the three imaging modalities: X-ray holography (top), scanning SAXS with nano-focused X-rays (center) and in situ STED microscopy (bottom). **b** Confocal (top left) and STED-image (lower right) of a cardiac tissue cell with labeled actin cytoskeleton. Scale bar: 5 μm. **c** 3D illustration of the sample in the STED (red) or the X-ray recording position (blue/green), respectively

array of 2D diffraction patterns. Each pattern corresponds to a local average of the diffraction signal over the illuminated area, given by the spot size of the KB-focus. In order to obtain clean diffraction patterns, parasitic scattering tails originating from the primary beam are removed by a set of soft edge apertures before the beam impinges on the sample. Further downstream, the primary beam is blocked by a beamstop, and the scattering cone is recorded by a 2D detector. In this setup, we use a Pilatus 300 k single photon counting detector (Dectris, Switzerland), which is placed on the same detector bench as the scintillation camera for X-ray holography. The pixel position is transformed to the scattering vector $|\mathbf{q}| = |\mathbf{k_{out}} - \mathbf{k_{in}}| = \frac{4\pi}{\lambda}\sin(\angle(\mathbf{k_{out}}, \mathbf{k_{in}})/2)$ as in

standard SAXS experiments, see also ref.[32] for the data processing scheme used here. The diffraction signal depends on the intrinsic structure factors of the molecular constituents reflecting the local short range order and orientation[25,28,30,31,56].

Depending on the signal level, contributions up to scattering vectors of about 2.3 nm$^{-1}$ have been recorded[25]. Note that scanning SAXS allows for various ways to compute contrast maps in the sample plane, depending on how the diffraction patterns are evaluated[30,32]. The method is especially sensitive to ordered structures, such as filament bundles of the cytoskeleton network. For example, the scan area can be mapped by integrating the diffraction intensity for each pixel, resulting in a so-called X-ray

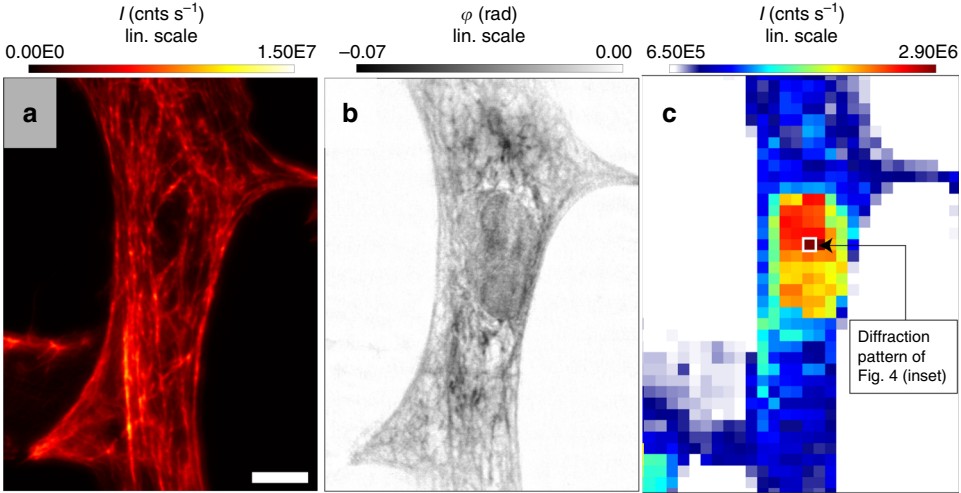

**Fig. 2** Single cell dataset in all three imaging modalities. **a** STED micrograph of a neonatal cardiac tissue cell, centered around the nucleus. Scale bar: 5 μm. **b** X-ray phase reconstruction. **c** X-ray dark field map of the cell obtained by scanning SAXS

dark field map, see Fig. 2c. As in conventional SAXS, data analysis requires suitable models, or is limited to an empirical description in terms of scattering cross sections, scattering exponents describing the power-law decay of the scattering intensity with **q**, as well as the principal direction of the scattering distribution and the corresponding degree of anisotropy[31].

For STED microscopy (Fig. 1a, bottom), the optical axis of the STED unit is aligned anti-parallel to the X-ray beam and is in close proximity to the X-ray recording position of the sample, see also Supplementary Note 1 and in particular Supplementary Fig. 1. The microscope consists of an excitation and a depletion (i.e., STED) laser, here with wavelengths of 632 nm (pulsed mode) and 775 nm, respectively. The beams are focused by a commercially available objective (UPLSAPO ×40 air objective, Olympus) onto the sample. The excitation laser reaches the sample in a Gaussian shape intensity profile. The depletion laser is shaped by a set of waveplates exploiting polarization properties of the beam (so-called easySTED-principle[57]), resulting in a doughnut-like intensity profile in the focal plane. The STED pulse impinges on the sample with a small time delay on the order of hundreds of picoseconds with respect to the excitation pulse. It depletes the fluorescent signal by stimulated emission in the entire annulus around the central zero (see bottom box, inset). Only the fluorescent signal from the central zero is recorded with single photon sensitivity by an avalanche photodiode (APD). This principle results in an effective probing spot significantly below the diffraction limit. In the present case, the beam traverses a small air gap (of here 180 μm working distance given by the objective) which is necessary in order to be able to laterally move the sample between the STED and X-ray positions. This constrains the numerical aperture, here to NA = 0.95, compromising resolution. However, the relative resolution advantage over confocal microscopy is maintained for a given numerical aperture, and significantly more features of the actin cytoskeleton can be visualized by STED when compared to a confocal scan, see Fig. 1b. As in other STED microscopes, confocal recordings are used as controls alongside with the STED recording. In addition, bright field and epifluorescence images can be acquired in a full-field mode of the microscope. This is especially useful for inspection of larger areas during initial alignment, the re-identification of particular sample areas and, above all, the finding of a suitable scan region. The design of the STED add-on to the GINIX endstation enables a quick switch between all three modes of the instrument: Immediately after the STED recording,

the sample tower can be shuttled towards the X-ray position using its motorized stage, see also Fig. 1c.

Data analysis: Holographic phase retrieval was implemented by first performing a contrast-transfer-function (CTF) step to estimate the sample support, which is then followed by iterative phase retrieval based on the support constraint and the constraint of a pure phase object[46,58]. These sample constraints, as well as the projection onto the measured hologram (magnitude constraint), are iteratively enforced using the update scheme of the relaxed averaged alternating reflections (RAAR) algorithm[59]. Parameters of geometry, CTF and RAAR algorithms, initialization, number of iterations and the definition of the sample support are given in the Supplementary Methods section, see in particular Supplementary Table 2. For scanning SAXS, the cellular signal, as recorded by a single photon counting pixel detector (Pilatus 300 k, Dectris), reaches a momentum transfer up to about 0.8 nm$^{-1}$. Each local diffraction pattern was summed up for the dark field signal, after masking out inter-module gaps and the beamstop area. Background subtraction, mask generation, and parameters of the automated SAXS analysis for computing the dark field and performing principal component analysis (PCA) are given in the Supplementary Methods section, see in particular Supplementary Fig. 3. For STED microscopy, the standard data acquisition provided by Abberior Instruments has been used, see also Supplementary Note 1.

**Multi-modal microscopy of cells**. We demonstrate the synergistic combination of the three imaging techniques on neonatal cardiac tissue cells. The cells adhering on glass slides are chemically fixed, and the actin cytoskeleton is labeled with Atto633 dye with an emission peak at $\lambda_{fl} = 651$ nm. The cells were prepared by plunge freezing followed by freeze-drying to keep the experiment simple, since X-ray and STED compatible cell chambers were not yet available, and also to circumvent the challenges associated with radiation damage and lower X-ray signal levels in the living state[27,28,30]. Figure 2 shows the results of a combined data acquisition on a single cell region that has first been pre-selected based on the bright field and epifluorescence micrographs in the full-field mode of the STED microscope. A coarsely scanned confocal overview of low radiation dose was then used to identify regions of interest (ROI), followed by an axial scan, which is used to find the correct focal plane with high precision. All STED images are recorded by a combined successive pixel-wise detection of the fluorescence signal in first

confocal and then in STED mode. The STED data, see Fig. 2a, resolves the labeled actin cytoskeleton of the cell beyond the diffraction limit, which in this case is $d_{\min.} = \lambda_{\mathrm{fl}}/(2\ \mathrm{NA}) = 651$ nm/1.9 ≈ 343 nm. After translating the sample towards the X-ray recording position, the ROI is re-identified by using a high magnification video camera. The camera is aligned on-axis with respect to the X-ray beam, with the X-ray beam position marked in its field of view from a previous calibration with a scintillation screen. After initial alignment of a first test sample or pattern, pre-stored values are available to shuttle between the STED and X-ray recording positions. Next, phase contrast X-ray images are recorded by X-ray holography, requiring only a relatively low radiation dose. The dose efficiency of X-ray holography compared to scanning SAXS or coherent diffractive imaging was recently demonstrated experimentally[46,60], and also illustrated by numerical simulations[61]. Figure 2b shows the holographic phase reconstruction as computed by an iterative algorithm based on support and phase object constraint[21], and using the RAAR update scheme[59]. Results exhibit filamentous structures along the main elongation axis of the cell. The filaments can partly be identified to be the same image components as in the corresponding STED micrograph of the actin cytoskeleton. In addition, unlabeled sub-structures are observed, notably the cell nucleus in the center. Finally, the scanning SAXS mode is applied, probing again the same ROI with the focused high flux KB beam. Figure 2c shows the resulting X-ray dark field of the selected scan region, as scanned with a step size of 1 μm. Although the spatial resolution in real space is much smaller in this image, which is primarily due to the comparably large step size and focal spot size, the diffraction signal comprises Fourier components up to high spatial frequencies. In this image we observe the strongest signal in the cell nucleus, and thus in the organelle which is absent in the STED micrograph of Fig. 2a. Based on the control of pixel size and scanning step size, all micrographs can be superimposed and registered, following numerical re-binning, see Supplementary Methods and in particular Supplementary Fig. 4.

**Correlative analysis of datasets**. Figure 3 shows the image analysis and segmentation used for comparing and correlating the data of the three different microscopy modalities. Segmentation of the actin filaments in the STED image as presented in Fig. 3a was performed using the so-called filament sensor algorithm introduced in ref. [62] (see Supplementary Methods and in particular Supplementary Table 1 for details). The corresponding segmentation of the holographic phase map of Fig. 2b is shown in Fig. 3b. Comparison of results of the two modalities indicates a largely consistent orientation for many filaments (see Supplementary Methods and in particular Supplementary Fig. 4 for quantification), in line with the polarization of the cell (i.e., its main elongation axis). We can infer that the actin filaments, which can be traced best from the STED data due to the specific labeling, correlate to a significant extent with the filaments segmented in the holographic X-ray image. In other words, actin filaments or filament bundles also stand out in electron density contrast, as can be recognized in particular at the top and bottom part of the cell of Fig. 2a, b. Figure 3c shows the pseudo-vector field (green lines) indicating the local direction (angle $\theta(y, z)$), along which molecular structures are preferentially oriented, as well as the parameter $\omega(y, z)$ (gray scale), which quantifies the anisotropy of the diffraction pattern for each scan point $(y, z)$. $\theta(y, z)$, and $\omega(y, z)$ are calculated by principal component analysis (PCA), yielding two principal directions, i.e. the direction of highest and lowest variance $\lambda_{1,2}$ of the momentum transfer, and the degree of anisotropy defined as $\omega_{\mathrm{pa}} = |\lambda_1 - \lambda_2|/(\lambda_1 + \lambda_2)$[31], see

also Supplementary Methods for details. Values for $\theta$ and $\omega$ are shown only if the integrated scattering intensity exceeds a threshold of $I_{\mathrm{df}} = 7.5 \cdot 10^5$ ph./s. In this manner, the analysis is limited to the cell only, excluding background. The graphs show that the anisotropy of the diffraction pattern is influenced by the orientation of actin filaments, see Fig. 3c.

This can be taken as an indication that the local polarization of the actin cytoskeleton acts as a director for many molecular components, lifting the symmetry with respect to the orientational degrees of freedom. Finally, Fig. 3d shows the phase map of the holographic reconstruction superimposed with the segmentation of the actin filaments from the STED recording, supporting the conclusion that thicker actin filaments also stand out in electron density.

**Range of spatial frequencies and resolution**. Next, we address the range of Fourier components which can be covered by each of the modalities, as well as their associated resolution limits. To this end, we have computed the power spectral densities (PSD) of the confocal, STED and holographic images, as well as the structure factor of the SAXS data. Note that the PSD is directly related to the signal and can be computed for all modalities, in contrast to an instrument-dependent optical transfer function (OTF), which is not a useful quantity for a diffraction experiment, unless the resolution is limited by wavelength or detector angle. Figure 4 shows their corresponding azimuthal averages, i.e., the one-dimensional PSD curves of the confocal image (yellow), the STED-image (red), the holographic reconstruction (blue), as well as the factor decay of the background corrected SAXS data from within the cell nucleus (green). For a direct comparison, all spatial frequencies $\nu$ from the PSDs are transformed into a scattering vector by $q = 2\pi\nu$. In order to determine the cross-over to the noise floor, power law functions of the form $I(q_{\mathrm{r}}) = a q_{\mathrm{r}}^b + c$ have been fitted to the slopes of the signal (bold gray lines). The cross-over points $q_i$ (with i attributed to confocal, STED, holography or scanning SAXS) are derived from $a q_i^b = bgr$, with $bgr$ denoting the constant background level, fitted at high $q$-values (dashed thin gray lines). Features smaller than the critical structure size $d_{\mathrm{cr}} = \pi/q_i$ must be attributed to noise.

As expected, results indicate a significant improvement in resolution from confocal to STED and to holographic X-ray imaging. Of course, the resolution in STED mode can be further optimized by using higher NA-objectives and immersion oil. This could, however, compromise the flexibility of the combined modalities, since the residual oil on the object could easily spoil the X-ray results, in particular if no homogeneous layer could be achieved.

The resolution of the X-ray holographic method could be further increased by using other waveguides or focusing optics also with higher NA, thus realizing a higher divergence behind the focal plane. More issues to be addressed are the vibration isolation, as well as detector efficiency and detector point spread function (PSF). With regard to scanning SAXS, the dynamic range of momentum transfer has to be increased. Currently, the smallest momentum transfer $q_{\mathrm{r,min}}$ is determined by the size of the beamstop, and the largest momentum transfer $q_{\mathrm{r,max}}$ by the detector module size. Improvements are easily possible by using pixel detectors with a higher number of modules, more precisely the EigerX 4M detector (Dectris, Switzerland) which will be made available for the next experiment, as well as increasing the acquisition time.

**Discussion**
The primary goal of this work is to identify and realize the simplest possible configuration in terms of instrumentation,

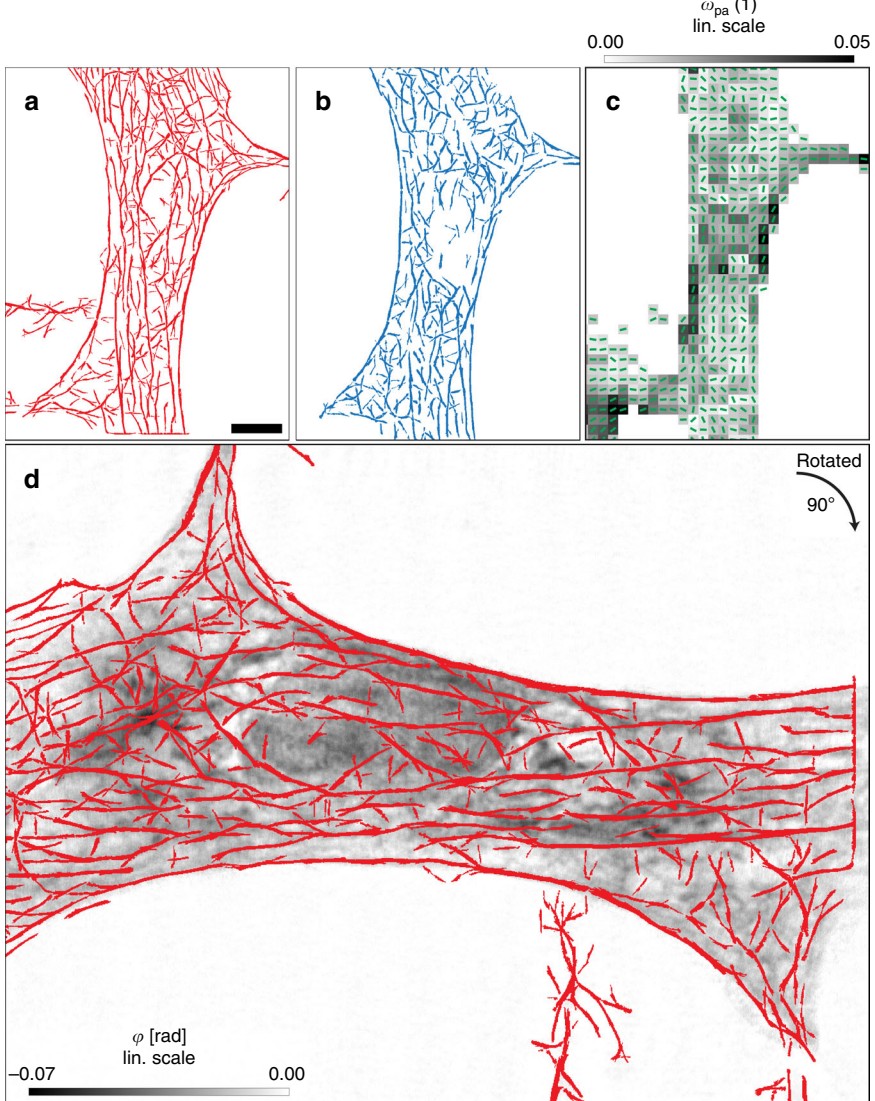

**Fig. 3** Filament analysis of all three contrast schemes. **a** Actin filaments detected in the STED micrograph by a filament sensor algorithm (red). Scale bar: 5 μm. **b** Filaments detected in the X-ray phase map (X-ray holography), using the same filament sensor algorithm (blue). **c** Analysis of the SAXS signal, yielding the anisotropy parameter $\omega$ (gray scale), which quantifies the degree of anisotropy of the diffraction patterns, as well as the direction of the principal scattering axis $\theta$ (lines correspond to long axis in real space), as obtained by PCA analysis of the diffraction pattern. Both $\omega$ and $\theta$ are thresholded to X-ray darkfield values above $I_{df} = 7.5 \cdot 10^5$ ph./s. **d** Superposition of STED filaments (red) and the phase map (gray scale) of the holographic reconstruction. The reconstructed phase is proportional to the projected electron density

sample preparation, as well as the choice of substrate to meet the STED and X-ray requirements in this first proof-of-principle experiment. For this reason, we have also chosen actin as a well-studied cytoskeletal component in neonatal cardiac tissue cells from the rat. To avoid complications in the design of X-ray and STED compatible chambers, which is possible in principle but not yet readily available, cells are freeze-dried after fluorescent labeling and fixation. Future extensions must be directed towards more relevant sample states and environments. This is facilitated by the fact that the present, quite flexible STED setup allows for adaptation with regards to the working distance of the objective, as well as the optical path. The results obtained demonstrate that the combination of STED microscopy, X-ray holographic phase contrast imaging and scanning SAXS with nano-focused radiation can be implemented at a synchrotron beamline in a straightforward manner. A direct correlation of data can be achieved by superimposing the fluorescence data of the STED micrograph, the phase reconstruction and the X-ray dark field

map. By registry to common structures such as the actin filaments, or more simply the overall contour of cells, the translational shifts between the datasets can be determined to allow for correlative interrogation of the data. These shifts can be determined with even more accuracy in future projects by applying an appropriate test pattern that can be visualized by all contrast schemes. The measured values of the motor positions can then be pre-stored and applied to all samples in an automated manner, and with a precision of down to tens of nanometers.

The fact that the actin network can at least be partially visualized in the X-ray phase reconstruction shows that filament bundles also stand out in local electron density against other cellular components. Furthermore, we can infer that the actin filament orientation is indeed correlated with the principal directions of the local SAXS pattern. This is in line with our earlier study, which was carried out at lower resolution with an offline epifluorescence microscope[30]. The combined approach of X-ray and STED microscopy already in this simple example

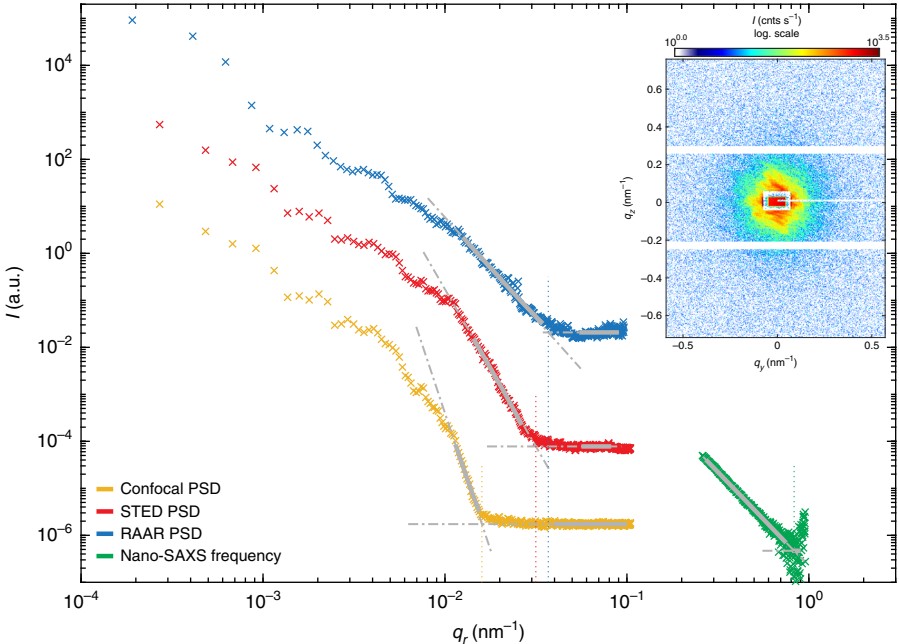

**Fig. 4** Power spectral densities (PSDs) of the confocal (yellow) and STED micrograph (red), the phase reconstructed image of the X-ray holography experiment (blue), along with the SAXS structure factor (power-law decay), all shown as a function of the scattering vector $q_r$. For a direct comparison, all PSD values are transformed into a $q$-equivalent by rescaling the x-axis with a factor of $2\pi$. The cross-over to the noise floor is determined to $q_i = 0.016$, 0.032, 0.037, and 0.83 nm$^{-1}$, for confocal, STED, holography and scanning SAXS data, respectively. Intensity values are shifted for clarity. The inset shows a typical diffraction pattern recorded in the nucleic region

shows that the combined information is more than the sum of the two: It may also help, for example, to constrain SAXS models and facilitate interpretation of the X-ray data. Moreover, the fact that the STED micrographs are probed directly within the synchrotron endstation will enable the further optimization of the X-ray recording parameters, and to closely monitor radiation damage of one or another X-ray experiment.

The next step is a significant improvement in instrumentation and workflow. Resolution and image quality of all three modalities can be enhanced, while maintaining the combined and correlative approach. Possible starting points in this respect are the implementation of STED objectives with higher NA, the testing of different substrates or carrier foils, an improved vibration isolation of all setup components involved, as well as further upgrades of waveguide optics, detection panels and positioning. Moreover, the recording and correlation of larger datasets will provide sufficient sampling of cells in view of statistical relevance. For this purpose, holographic full-field recordings will be combined with sample translations and stitching of images to cover many different cells up to the entire sample substrates. To obtain maps of chemical composition, the correlative imaging approach could also be easily extended by recording the X-ray fluorescence along with the SAXS scans.

A medium-term goal is to extend the correlative imaging approach to hydrated chemically fixed, and finally also to living cells. This will circumvent the structural degradation by the preparation process, which is without a doubt considerable for the freeze-drying process. With a suitable configuration and design established, this will allow to apply physiologically meaningful conditions in the long-term, as well as make it possible to observe dynamic processes in chemistry or biology. At the same time, hydrated cells result in a significantly reduced X-ray signal, compared to the present sample, and radiation damage will become a major concern. In this scenario, the online configuration of the STED microscope during synchrotron measurement campaigns can serve as a much needed control in view

of the short timespan over which cells can detach, perform apoptosis or undergo other structural changes as a result of radiation damage. These important mechanisms can thus be studied in detail. Applications of the correlative imaging approach with a well controlled dose can address a variety of biological structures in cells and tissues. Cytoskeleton structure, as shown here, and mechanical functions such as the contractility in cardiomyocytes based on the underlying sarcomeric structure, can take advantage of correlative X-ray and STED microscopy. To this end, a stably transfected cardiac cell-line with a STED-compatible protein-based fluorescence dye attached to the target structure as in refs.[63,64]. would be extremely beneficial.

## Methods

**Cell source and cultivation**. Neonatal rat cardiac tissue cells were obtained from 1–2 days old neonatal Wistar rats. Ventricles were sliced in phosphate buffer into about 1 mm big sections, followed by 3 to 4 digestion steps with collagenase II (300 U ml$^{-1}$, Worthington, 15 min at 37 °C) and centrifugation, see also refs. [65,66]. A glass coverslip serving as the substrate (thickness #1) was plasma-cleaned for several minutes. It was then transferred to a 12-well plate under a clean bench and coated with fibronectin/gelatin-solution (F4759 and G7041, Sigma-Aldrich). The well was flooded with a cell suspension of $2 \times 10^5$ cells/ml in medium (DMEM-F12, 11039-021, Gibco; 10% FCS, 10270–106, Gibco; 1% pen./strep. 15140–122, Gibco) until the substrate was fully covered. The sample was then transferred into the cell incubator (37 °C, 5% $CO_2$) allowing the cells to settle on the substrate.

**Sample fixation, labeling, and freeze-drying**. The adhered cells were fixed the next day using 9% formaldehyde (252549, Sigma-Aldrich) in Dulbecco's Phosphate-Buffered Saline (DPBS D8537, Sigma-Aldrich). They were then permeabilized with Triton X-100 and the actin cytoskeleton was labeled with a solution of 0.5 μM Phalloidin-Atto633 (AD 633-8X, Atto-Tec) in DPBS following manufacturer protocols. Samples were plunged into an ultracool bath (tempered at about −195 °C) of a liquid ethane/propane mixture using a Leica GP EM grid plunging system. Samples were then transferred into a home-built lyophilizator and freeze-dried.

**Code availability**. Computer code for nanodiffraction analysis presented here is available on Github, see ref.[32] and https://irpgoe.github.io/nanodiffraction/, and from the corresponding author upon request.

## Data availability

Data and explanation supporting the findings of the here presented work are available on the article, its Supplementary Information file, in https://doi.org/10.5281/zenodo.1305080 and from the corresponding author upon request.

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

## Acknowledgements

We thank Stefan Hell for continuous support and advice; Susanne Hengst for help with cell culture; Matthias Meister, Peter Luley and Bastian Hartmann for engineering support; Benjamin Eltzner for software support with reference to the filament sensor; and Stefan Luther and Marion Kunze for providing heart tissue cells. We acknowledge SFB937 (A07 and A11) for funding the correlative microscopy of actin cytoskeleton networks, and BMBF/05K16MG2 for funding of the custom-built STED microscope.

## Author contributions

T.S. and S.K. designed research and are PIs of the corresponding project; H.M., M.R., and B.H. designed and built the STED microscope; M.O., M.B., and M.S. implemented the STED microscope into the GINIX-setup; M.B., J.D.N., M.O., A.W., and T.S. performed the synchrotron experiments; M.B. and J.D.N. analyzed data; M.B. and T.S. wrote the manuscript, with contributions from all authors.

## Additional information

**Competing interests:** The authors declare no competing interests.

