## [Peer Review File · Nature Communications]

Reviewers' comments:

Reviewer #1 (Remarks to the Author):

NCOMMS-18-10155

Combined hard x-ray and STED microscopy of cardiac tissue cells

The paper deals with a "correlative microscopy approach for biological cells based on holographic x-ray imaging, x-ray scanning diffraction and stimulated emission depletion (STED)-microscopy." The paper is interesting but, starting from the title, the results are not reported with enough care.

Title: the title is incomplete and unfocused. A more appropriate choice could be "Correlative Imaging of cardiac tissue by x-ray holography, x-ray scanning diffraction and stimulated emission depletion microscopy"

Abstract: the authors state that "both labeled biomolecules and unlabeled structures in the cell are visualized in a complementary manner, and can be recorded in a quasi-simultaneous scheme." The use of the word "quasi-simultaneous", which is a not scientific expression, sounds extremely imprecise. Either data are collected simultaneously or not. This is not a simultaneous data collection experiment, which eventually is the weakness of the approach considering that radiation damage may occur during the sequential data collection. Authors have to remove the word "quasi-simultaneous" through the paper.

Literature: this combined approach, based on the correlation of 3 different imaging techniques, is not a novel idea, and literature concerning previous papers in the same field are not properly cited. Here a short list of experiments dedicated to biological tissues, which should be added:

Correlative imaging of biological tissues with apertureless scanning near-field optical microscopy and confocal laser scanning microscopy

By: Stanciu, SG et al. BIOMEDICAL OPTICS EXPRESS

Volume: 8 Issue: 12 Pages: 5374-5383 Published: DEC 1 2017

Correlative Light and Scanning X-Ray Scattering Microscopy of Healthy and Pathologic Human Bone Sections

By: Giannini, C et al. SCIENTIFIC REPORTS

Volume: 2 Article Number: 435 Published: MAY 31 2012

Cryo X-ray microscope with flat sample geometry for correlative fluorescence and nanoscale tomographic imaging

By: Schneider, G et al. JOURNAL OF STRUCTURAL BIOLOGY

Volume: 177 Issue: 2 Pages: 212-223 Published: FEB 2012

Correlative light-electron microscopy (CLEM) combining live-cell imaging and immunolabeling of ultrathin cryosections

By: van Rijnsoever, C et al. NATURE METHODS

Volume: 5 Issue: 11 Pages: 973-980 Published: NOV 2008

Implementation of the STED-setup into a hard x-ray microscopy end-station: this section should be re-written describing the sequential procedure of operations adopted for data collection (clarity will improve very much with a bullet point list)

Data analysis is not exhaustively described. For example, concerning holography, the authors state in a very general and insufficient way that phase reconstruction is realized by "performing a contrast-transfer-function (CTF) step to estimate the sample support, which is then followed by iterative phase retrieval based on the support constraint and the constraint of a pure phase

object"; concerning scanning SAXS it is not clear if the entire scattering range of angles is used for the analysis or they selected a precise smaller range (length scale). The method section should be completed including paragraphs dedicated to the data analysis adopted for each of the three imaging techniques.

General remark: inaccuracies (such as "extend" instead of extent in the abstract; "helps" instead of help at the first line of page 2; "with with" instead of with in the paragraph Implementation of the STED-setup into a hard x-ray microscopy end-station; "benefical" instead of beneficial at the last line of the Discussion and Outlook; [57] is missing) reveal a certain hurry in the paper writing.

Major revisions are mandatory before a final decision is taken.

Reviewer #2 (Remarks to the Author):

The authors present a proof-of-principle study of quasi-simultaneous correlative x-ray and STED microscopy. Correlative microscopy is a growing field of utmost importance since the connection of complementary readouts reveals essential new details. The correlation of x-ray and STED microscopy is just what was missing and I am excited to see it working. The authors have chosen a very nice and clever approach by reproducibly sliding the sample between the x-ray and STED microscopy detection lines, and the approach is verified by imaging and correlating imaging data of actin in plunge frozen fixed cells. The data is convincing, the approach excellent and the manuscript well written. I can only strongly suggest publication of this important and exciting work in Nature Communications.

I still have some minor comments which should however be approached prior to publication – the authors will be able to straightforwardly handle those.

1) I am missing supplementary information describing the essential details of the experimental realization (e.g. STED setup, overlay protocol for catching the same ROI between the three different approaches and its reproducibility, first test samples and patterns, sample preparation, ...) and data analysis (numerical re-binning, phase retrieval, CTF, segmentation algorithm, RAAR, PCA, background-correction, missing data relief, ...) - I know some of them (but only some) have been introduced in references given, but the experimental and analysis details should still be briefly repeated.

2) I also could in parts not follow certain details – I suggest an improved description:

- Page 4, middle: What is meant by dose efficiency here?

- Page 4, bottom: Specify/name step size and focal spot size.

- Page 5, middle: Quantify consistency and correlation of actin structures between STED and holographic X-ray imaging (e.g. parameters such as Pearson coefficient) – "to a significant extent" is a little bit imprecise.

- Page 5, middle: What does "principal axes of anisotropy" mean – not clear here?

- Page 5, bottom: The orientation and ordering parameters have to be explained in a better way – hard to follow. Why has this specific scattering intensity threshold been chosen? I did not understand the sentence starting with "Note that this mapping ...". Which "phase map" is meant in the second last line and which conclusions again (last line, could be repeated to keep the flow)?

- Page 6, top and figure caption: The power spectral density (PSD) and scattering vector are tough to understand – how is they related to OTF (which is usually used in optical microscopy)? I did not understand the transformation into q-equivalent. How were the cross-over values (to the noise level) determined/defined?

Page 6, top: Why has the SAXS-related analysis been done within the cell nucleus – the results

must severely depend on the chosen area?

Page 6, bottom: What is meant by PSF and nanodiffraction here?

Page 6, bottom: Why has the acquisition time not been increased already in the current measurements?

Page 7, top: What is meant by "nano-focused"?

Page 7, middle: I did not really find (in the results section) how the translational shifts had been determined?

3) References: [3] does not report on STED. [57] is empty.

Reviewer #3 (Remarks to the Author):

Correlative microscopy that combines different imaging modalities provides a powerful tool to study sample in a more comprehensive way. This seems to be a trend for future microscopes that integrate multi-modal approaches from a same source and/or different sources (optical, X-ray, electron, etc) into one instrument and study the sample in a same environment. This correlative imaging idea has been demonstrated in many ways, such as Correlative Light and Electron Microscopy (CLEM) [1] (review article) and correlative light and soft X-ray cryo-microscopy [2], from this perspective the novelty of this manuscript may not be very high. However, this manuscript does firstly demonstrate a good marriage between STED super-resolution light microscopy and high-resolution X-ray imaging (two most powerful techniques in light and x-ray realms, respectively), which may open up new opportunities in the microscopy community. The following are my comments:

1. The key point of this paper is correlative imaging by combining STED microscopy and X-ray imaging, so I think authors should provide more information in the introduction about correlative microscopy and relevant references. In addition, Ref. 25 was not cited properly, along with one published earlier [3] they are about correlative imaging between X-ray fluorescence and ptychography, not just X-ray fluorescence.
2. Due to different imaging mechanisms, all three imaging modalities were conducted in sequence, not simultaneously, so I would rather not call this "quasi-simultaneous scheme". Instead, it is better to mention that all three imaging modalities were obtained from almost the same sample environment rather than other correlative methods which obtained different modalities in different setups. Regarding to simultaneous imaging, why didn't authors add a fluorescence detector in the scanning SAXS setup to obtain elemental maps?
3. The bottom-left part of SAXS image in Fig. 3c is not consistent with the other two imaging modalities. Can authors explain on this? What is step size in the scanning SAXS? The diffraction patterns from SAXS can provide high-resolution information from sample, however, the spatial resolution of its dark field map is poor due to the large scan step size. Since SAXS measurement is similar to ptychography, is it possible to get ptychographic image with high spatial resolution if there is enough overlap between scan points and spatial coherence?
4. What is the spot size (or the remaining area allowing fluorescence) of STED microscopy?
5. Authors claimed that the STED image in Fig. 3a and holographic phase image in Fig. 3b have good consistence in the orientation for many filaments. Is there a better way to show this? For example, since Fig. 3b includes both labeled and unlabeled structures, can those filaments that are consistent with those in Fig. 3a be labeled with different color?

Minor comments:

1. The geometrical magnification M in X-ray holography is incorrect, it should be the reciprocal of the current form.
2. Double check the unit of the scale bar in Fig. 2b and Fig. 3d is degree or radian? It seems to me the phase change through the cell is too small if it is in degrees.
3. In page 4 line 3, i.e. STED- \diamond STED
4. In page 4, "...the challenges associates with with ...", delete one "with"

References:

- [1] Briegel et al., Correlated light and electron cryo-microscopy, *Methods Enzymol.* 481, 317 (2010)
- [2] Hagen, et al., Correlative VIS-fluorescence and soft X-ray cryo-microscopy/tomography of adherent cells, *Journal of Structural Biology* 177, 193(2012)
- [3] Deng, et al., Simultaneous cryo x-ray ptychographic and fluorescence microscopy of green algae, *PNAS* 112, 2314(2015)

Manuscript NCOMMS-18-10155: Reply to Reviewers

Comments/Reply Reviewer #1:

Combined hard x-ray and STED microscopy of cardiac tissue cells

1. The paper deals with a “correlative microscopy approach for biological cells based on holographic x-ray imaging, x-ray scanning diffraction and stimulated emission depletion (STED)-microscopy.” The paper is interesting but, starting from the title, the results are not reported with enough care.

Reply: We are glad that the reviewer considers our work interesting, and have improved clarity and presentation, by rephrasing different paragraphs of the main article. Moreover, we have added a supplementary information section to this work in order to describe key aspects of the work in full detail. We thank the reviewer for his/her helpful advice.

2. Title: the title is incomplete and unfocused. A more appropriate choice could be “Correlative Imaging of cardiac tissue by x-ray holography, x-ray scanning diffraction and stimulated emission depletion microscopy”

Reply: We have changed the title following this suggestion.

3. Abstract: the authors state that “both labeled biomolecules and unlabeled structures in the cell are visualized in a complementary manner, and can be recorded in a quasi-simultaneous scheme.” The use of the word “quasi-simultaneous”, which is a not scientific expression, sounds extremely imprecise. Either data are collected simultaneously or not. This is not a simultaneous data collection experiment, which eventually is the weakness of the approach considering that radiation damage may occur during the sequential data collection. Authors have to remove the word “quasi-simultaneous” through the paper.

Reply: We have replaced "quasi-simultaneous", by 'one after the other'. Importantly, the two modalities can be carried out using the same environment and mounting, and within a short time delay.

4. Literature: this combined approach, based on the correlation of 3 different imaging techniques, is not a novel idea, and literature concerning previous papers in the same field are not properly cited. Here a short list of experiments dedicated to biological tissues, which should be added:

- Correlative imaging of biological tissues with apertureless scanning near-field optical microscopy and confocal laser scanning microscopy
By: Stanciu, SG et al. BIOMEDICAL OPTICS EXPRESS
Volume: 8 Issue: 12 Pages: 5374-5383 Published: DEC 1 2017
- Correlative Light and Scanning X-Ray Scattering Microscopy of Healthy and Pathologic Human Bone Sections
By: Giannini, C et al. SCIENTIFIC REPORTS
Volume: 2 Article Number: 435 Published: MAY 31 2012
- Cryo X-ray microscope with flat sample geometry for correlative fluorescence and nanoscale tomographic imaging
Correlative light-electron microscopy (CLEM) combining live-cell imaging and immunolabeling of ultrathin cryosections
By: van Rijnsoever, C et al. NATURE METHODS
Volume: 5 Issue: 11 Pages: 973-980 Published: NOV 2008

Reply: We did not mean to convey the idea that this is the first correlative x-ray and optical microscopy experiment. We have narrowed the statement highlighting nanoscopy and STED, and in the introduction and have include the three references, as well as two more references to correlative microscopy, pointed out by reviewer 3.

5. Implementation of the STED-setup into a hard x-ray microscopy end-station: this section should be rewritten describing the sequential procedure of operations adopted for data collection (clarity will improve very much with a bullet point list)

Reply: We have rephrased the paragraph and now specify the exact time sequence of the recordings. We have also included more details on the instrumentation with an additional figure as supplementary information.

6. Data analysis is not exhaustively described. For example, concerning holography, the authors state in a very general and insufficient way that phase reconstruction is realized by “performing a contrast-transfer-function (CTF) step to estimate the sample support, which is then followed by iterative phase retrieval based on the support constraint and the constraint of a pure phase object”; concerning scanning SAXS it is not clear if the entire scattering range of angles is used for the analysis or they selected a precise smaller range (length scale). The method section should be completed including paragraphs dedicated to the data analysis adopted for each of the three imaging techniques.

Reply: We have added a section on data analysis in the section describing the implementation, as suggested. Further, we have added an even more detailed description with all parameters used in the data analysis as supplementary information. In particular, the exact procedures for all three modalities are now described, as requested.

7. General remark: inaccuracies (such as “extend” instead of extent in the abstract; “helps” instead of help at the first line of page 2; “with with” instead of with in the paragraph Implementation of the STED-setup into a hard x-ray microscopy end-station; “benefical” instead of beneficial at the last line of the Discussion and Outlook; [57] is missing) reveal a certain hurry in the paper writing.

Reply: We apologize for the typos, and have carefully corrected all mistakes and proof-read the entire article again, in view of both spelling and English grammar.

We thank the reviewer for these corrections and recommendations to improve the manuscript.

Comments/Reply Reviewer #2:

1. I am missing supplementary information describing the essential details of the experimental realization (e.g. STED setup, overlay protocol for catching the same ROI between the three different approaches and its reproducibility, first test samples and patterns, sample preparation, ...) and data analysis (numerical rebinning, phase retrieval, CTF, segmentation algorithm, RAAR, PCA, background-correction, missing data relief, ...) - I know some of them (but only some) have been introduced in references given, but the experimental and analysis details should still be briefly repeated.

Reply: In response to this suggestion, as well as that of reviewer 1, we now provide supplementary information (SI) on all aspects of the experiment and data analysis.

2. I also could in parts not follow certain details – I suggest an improved description:

(i) Page 4, middle: What is meant by dose efficiency here?

(ii) Page 4, bottom: Specify/name step size and focal spot size.

(iii) Page 5, middle: Quantify consistency and correlation of actin structures between STED and holographic X-ray imaging (e.g. parameters such as Pearson coefficient) – “to a significant extent” is a little bit imprecise.

(iv) Page 5, middle: What does “principal axes of anisotropy” mean – not clear here?

(v) Page 5, bottom: The orientation and ordering parameters have to be explained in a better way – hard to conclusions again (last line, could be repeated to keep the flow)?

(vi) Page 6, top and figure caption: The power spectral density (PSD) and scattering vector are tough to understand – how is they related to OTF (which is usually used in optical microscopy)? I did not understand the transformation into q-equivalent. How were the cross-over values (to the noise level) determined/defined?

(vii)Page 6, top: Why has the SAXS-related analysis been done within the cell nucleus – the results must severely depend on the chosen area?

(viii)Page 6, bottom: What is meant by PSF and nanodiffraction here?

(ix) Page 6, bottom: Why has the acquisition time not been increased already in the current measurements?

(x) Page 7, top: What is meant by “nano-focused”?

(xi) Page 7, middle: I did not really find (in the results section) how the translational shifts had been determined?

Reply:

- (i) Dose efficiency means that for given resolution, the reconstructed image can be obtained at lower dose. This issue is explained in ref.[47,55] from an experimental point of view, and in ref.[56] from a theoretical point of view, as we now explain more clearly.

- (ii) The step size was 1 μm , and the spot size 300nm, as we now specify in the text on p.4 describing Fig.2(c).
- (iii) We have now quantified the match of pixel counts, quantifying the simultaneous occurrence of filaments in Fig. 3(a) and (b), as shown in an additional figure in the SI.
- (iv) The entire paragraph has been reformulated to explain the concept of the principal axes
- (v) ...and of anisotropy, and the parameter ω . In addition, the SI gives details of the analysis.
- (vi) Since the 'numerical aperture' of the diffraction signal itself determines the resolution and not the instrument (detector size, wavelength), we do not recur to the OTF (Fourier transform of the PSF) of the instrument, but use the signal itself, or more precisely its power spectral density (PSD), as we now explain in more detail. The power spectral density (PSD) is defined as the modulus squared of the Fourier transform of the signal: signal is electron density $\rho(x,y)$, so that the PSD becomes equal to the scattering intensity $I \sim |\text{FT}[\rho]|^2$. To estimate the resolution, the intensity is averaged azimuthally, and plotted against q . When $I(q)$ levels off into a (white noise) plateau, this cross-over is a measure of the spatial resolution. Here, q is the spatial Fourier component times 2π (spatial angular frequency). All definitions and procedures are now fully specified in the SI.
- (vii) We have selected the cell nucleus, since the SAXS signal was particularly strong in this organelle, and since this organelle has not been labelled, so that the SAXS indeed reflects the non-labelled structure.
- (viii) We have now clarified that we meant *detector point spread function*. The word "nano-diffraction" denotes the scanning SAXS experiments with nano-focused beams. The word has been replaced by "scanning SAXS" for the sake of an identical notation throughout the MS.
- (ix) Indeed, the acquisition time could have been increased already for this experiment. However, in this first beamtime, we had chosen to scan multiple cells at the cost of shorter scans each.
- (x) If the probing x-ray beam is focused to below a micron, (for example by KB-mirrors) this is conventionally already designated as a nano-focused beam in synchrotron science. Here we have used a spot size measured to 324x309 nanometers. For clarity, we have added an explanation on page 2 behind the specification of the spot size.
- (xi) The translational shifts between all three datasets were adjusted 'by eye'. To this end, the holographic reconstruction of (b) and the x-ray dark field of (c) were rescaled using Matlab routines (*imresize*), so achieve equivalent pixel size. An additional figure in the SI illustrates the three combinations of datasets, with STED (green), holography (blue) and scanning SAXS (red).

3. References: [3] does not report on STED. [57] is empty.

Reply: The cited work of Balzarotti et al. reports on the "minflux" imaging method, which combines basic concepts of STORM- and STED-microscopy. Since it differs from the STED technology used here, we agree that we can omit it. Citation [57] now comprises an url, linking the article to the github-entry where the current version of the nanodiffraction toolbox can be downloaded.

We thank the reviewer for these questions and corrections which have helped us to improve the manuscript a lot.

Comments/Reply Reviewer #3:

1. (i) The key point of this paper is correlative imaging by combining STED microscopy and X-ray imaging, so I think authors should provide more information in the introduction about correlative microscopy and relevant references.

(ii) In addition, Ref. 25 was not cited properly, along with one published earlier [3] they are about correlative imaging between X-ray fluorescence and ptychography, not just X-ray fluorescence.

Reply:

- (i) We have reformulated the second paragraph of the introduction in order to sharpen the focus on correlative microscopy.

- (ii) We have rephrased the respective sentence, have included a further reference to x-ray fluorescence microscopy of cardiomyocyte cells by Palmer et al. (2006), and have put the work of Deng et al. into a clearer context.

2. Due to different imaging mechanisms, all three imaging modalities were conducted in sequence, not simultaneously, so I would rather not call this “quasi-simultaneous scheme”. Instead, it is better to mention that all three imaging modalities were obtained from almost the same sample environment rather than other correlative methods which obtained different modalities in different setups. Regarding to simultaneous imaging, why didn't authors add a fluorescence detector in the scanning SAXS setup to obtain elemental maps?

Reply: We have replaced "quasi-simultaneous", by 'one after the other' (see also Reply to Reviewer#1/Q3). As pointed out/suggested by the reviewer, we now stress that the two modalities can be carried out using the same environment and mounting, and within a short time delay. Concerning x-ray fluorescence, we simply did not have an efficient (fast) implementation of a fluorescence detector at hand for the first experiment, but this is a very worthwhile perspective for the future, as we now mention in the outlook.

3. (i) The bottom-left part of SAXS image in Fig. 3c is not consistent with the other two imaging modalities. Can authors explain on this?

(ii) What is step size in the scanning SAXS?

(iii) The diffraction patterns from SAXS can provide high-resolution information from sample, however, the spatial resolution of its dark field map is poor due to the large scan step size. Since SAXS measurement is similar to ptychography, is it possible to get ptychographic image with high spatial resolution if there is enough overlap between scan points and spatial coherence?

Reply:

(i) The discrepancy in the bottom-left between the scanning SAXS in (c) and the segmented holographic reconstruction in (b) is due to the chosen support for the holographic reconstruction and segmentation. In fact, the DIC-micrograph of the respective area below reveals an adjacent cell in the bottom-left part. Obviously the SAXS signal of this cellular extension is relatively strong (x-ray darkfield), while the concentration of actin filaments must be quite low.

(ii) The scanning stepsize was 1 μ m, as we now mention in the MS.

(iii) We agree that this in principle a good idea. In fact, in Wilke et al. (2013) and Clement et al. (2016), we have used combinations of SAXS and ptychography. Recordings at the same time (same scan), however, were always difficult due to the necessity of a beamsptop (for SAXS), and the associated difficulty for ptychographic reconstruction. Further, SAXS results can be interpreted in meaningful ways, even for thick cells, while the exit wave of a single projection alone (without tomographic ptychography) is often less informative.

Clement et al. (2016): C. Y. J. Hémonnot, J. Reinhardt, O. Saldanha, J. Patommel, R. Graceffa, B. Weinhausen, M. Burghammer, C. G. Schroer, and S. Köster. X-rays Reveal the Internal Structure of Keratin Bundles in Whole Cells. ACS Nano 2016 10 (3), 3553-3561.

4. What is the spot size (or the remaining area allowing fluorescence) of STED microscopy?

Reply:

- 30nm step size for result shown in Fig. 2 (a), as is now mentioned.

- The calculated resolution (remaining area) for the STED with given objective (air, NA=0.95) is 90nm at full power. Using DNA based test structures (nanorulers, Gattaquant) a 140nm spacing could be resolved.

5. Authors claimed that the STED image in Fig. 3a and holographic phase image in Fig. 3b have good consistence in the orientation for many filaments. Is there a better way to show this? For example, since

Fig. 3b includes both labeled and unlabeled structures, can those filaments that are consistent with those in Fig. 3a be labeled with different color?

Reply: We have now color labeled consistent pixels in an additional overlay picture included as SI, and have quantified the matching pixel counts, see also Reply to Reviewer#2/2(ii).

Minor comments

1. The geometrical magnification M in X-ray holography is incorrect, it should be the reciprocal of the current form. Reply: has been corrected
2. Double check the unit of the scale bar in Fig. 2b and Fig. 3d is degree or radian? It seems to me the phase change through the cell is too small if it is in degrees. Reply: It is indeed radians. Has been corrected
3. In page 4 line 3, i.e. STED- à STED Reply: has been corrected
4. In page 4, "...the challenges associates with with ...", delete one "with" Reply: has been corrected

We apologize for these mistakes and are very grateful for the careful reading of the reviewer.

[1] Briegel et al., Correlated light and electron cryo-microscopy, Methods Enzymol. 481, 317 (2010)

[2] Hagen, et al., Correlative VIS-fluorescence and soft X-ray cryo-microscopy/tomography of adherent cells, Journal of Structural Biology 177, 193(2012)

[3] Deng, et al., Simultaneous cryo x-ray ptychographic and fluorescence microscopy of green algae, PNAS 112, 2314(2015)

Reply: We thank the reviewer for pointing out these relevant publications, which we have now included in the referenced work, and for all suggestions/corrections which have been of significant value and help.

REVIEWERS' COMMENTS:

Reviewer #1 (Remarks to the Author):

The authors implemented all corrections, in the manuscript and supplementary information, apart for the literature list.

In my previous review, I wrote:

Literature: this combined approach, based on the correlation of 3 different imaging techniques, is not a novel idea, and literature concerning previous papers in the same field are not properly cited. Here a short

list of experiments dedicated to biological tissues, which should be added:

>> Correlative imaging of biological tissues with apertureless scanning near-field optical microscopy

and confocal laser scanning microscopy

By: Stanciu, SG et al. BIOMEDICAL OPTICS EXPRESS

Volume: 8 Issue: 12 Pages: 5374-5383 Published: DEC 1 2017

>> Correlative Light and Scanning X-Ray Scattering Microscopy of Healthy and Pathologic Human Bone Sections

By: Giannini, C et al. SCIENTIFIC REPORTS

Volume: 2 Article Number: 435 Published: MAY 31 2012

>> Cryo X-ray microscope with flat sample geometry for correlative fluorescence and nanoscale tomographic imaging

Correlative light-electron microscopy (CLEM) combining live-cell imaging and immunolabeling of ultrathin cryosections

By: van Rijnsoever, C et al. NATURE METHODS

Volume: 5 Issue: 11 Pages: 973-980 Published: NOV 2008

The authors replied: We did not mean to convey the idea that this is the first correlative x-ray and optical microscopy

experiment. We have narrowed the statement highlighting nanoscopy and STED, and in the introduction

and have include the three references, as well as two more references to correlative microscopy, pointed

out by reviewer 3.

Unfortunately, none of the references suggested by reviewer 1 and reviewer 3 are present in the actual list of references.

This correction has been neglected. The manuscript should be revised accordingly.

Reviewer #2 (Remarks to the Author):

The authors have well commented on all of my (and in my opinion the other referees') concerns and accordingly revised the manuscript. It reads very well. I suggest publication of this excellent piece of work as is.

Reviewer #3 (Remarks to the Author):

I am mostly satisfied with authors' reply and the revised manuscript. I think the manuscript can be accepted for publication after minor revision.

Please follow journal's requirements for the reference format. For example, "et al." appears in a weird place in Ref. [4], [9], [11], [33], [35], [43]. In [6], "science" should be capitalized.

In addition, here are some comments for Supplementary Information:

1. The figure labels in the supplementary information (SI) are the same as these in the main text, which makes confusion. Please change those in SI, for example, Supplementary Figure 1.
2. Page 1: change "signal" to "signals" in "The signal from two contact switches at the sides are fed back to"
3. Page 1: there is a typo in this sentence "...with an average STED-Laser power of up to 1.25W = 1W." Please correct it.
4. Comparing the parameters between in Table 1, should the first "Sigma" in RAAR reconstruction be "Factor"?
5. Page 3: "...between recording of the cell and the empty beam." , change "recording" to "recordings".
6. The caption of Supplementary Fig. 3 is not right. The description for (a) is missing. Both the paragraph above the figure and the (c) caption mention "the radial binary mask (black frame)" or "dark field evaluation mask as black lines", however, this mask is not shown in the (c) figure.
7. Page 4, "Figure" can be abbreviated as "Fig." except at the start of a sentence. Please change "Fig. 3 shows ..." to "Figure 3 shows ..." or "Supplementary Fig. 3 shows ...".

Manuscript NCOMMS-18-10155B: Reply, 3rd Revision

Comments/Reply Reviewer #1:

The authors implemented all corrections, in the manuscript and supplementary information, apart for the literature list. In my previous review, I wrote:

Literature: this combined approach, based on the correlation of 3 different imaging techniques, is not a novel idea, and literature concerning previous papers in the same field are not properly cited. Here a short list of experiments dedicated to biological tissues, which should be added:

>> Correlative imaging of biological tissues with apertureless scanning near-field optical microscopy and confocal laser scanning microscopy

By: Stanciu, SG et al. BIOMEDICAL OPTICS EXPRESS

Volume: 8 Issue: 12 Pages: 5374-5383 Published: DEC 1 2017

>> Correlative Light and Scanning X-Ray Scattering Microscopy of Healthy and Pathologic Human Bone Sections

By: Giannini, C et al. SCIENTIFIC REPORTS

Volume: 2 Article Number: 435 Published: MAY 31 2012

>> Cryo X-ray microscope with flat sample geometry for correlative fluorescence and nanoscale tomographic imaging

Correlative light-electron microscopy (CLEM) combining live-cell imaging and immunolabeling of ultrathin cryosections

By: van Rijnsoever, C et al. NATURE METHODS

Volume: 5 Issue: 11 Pages: 973-980 Published: NOV 2008

The authors replied: We did not mean to convey the idea that this is the first correlative x-ray and optical microscopy experiment. We have narrowed the statement highlighting nanoscopy and STED, and in the introduction and have include the three references, as well as two more references to correlative microscopy, pointed out by reviewer 3. Unfortunately, none of the references suggested by reviewer 1 and reviewer 3 are present in the actual list of references. This correction has been neglected. The manuscript should be revised accordingly.

Reply: We have now successfully implemented the citations suggested from reviewer #1 and #3. In the previous version, we have already adapted the manuscript accordingly (Latex & Jabref), but accidentally overlooked a compilation error in the process to create the output file, so that a small passage including the citations did not appear in the corresponding pdf. We apologize for this mistake!

Comments/Reply Reviewer #2:

The authors have well commented on all of my (and in my opinion the other referees') concerns and accordingly revised the manuscript. It reads very well. I suggest publication of this excellent piece of work as is.

Reply: We are delighted to hear these nice words.

Comments/Reply Reviewer #3:

I am mostly satisfied with authors' reply and the revised manuscript. I think the manuscript can be accepted for publication after minor revision.

Please follow journal's requirements for the reference format. For example, "et al." appears in a weird place in Ref. [4], [9], [11], [33], [35], [43]. In [6], "science" should be capitalized.

In addition, here are some comments for Supplementary Information:

1. The figure labels in the supplementary information (SI) are the same as these in the main text, which makes confusion. Please change those in SI, for example, Supplementary Figure 1.
2. Page 1: change "signal" to "signals" in "The signal from two contact switches at the sides are fed back to"
3. Page 1: there is a typo in this sentence "...with an average STED-Laser power of up to 1.25W = 1W." Please correct it.
4. Comparing the parameters between in Table 1, should the first "Sigma" in RAAR reconstruction be "Factor"?
5. Page 3: "...between recording of the cell and the empty beam." , change "recording" to "recordings".

6. The caption of Supplementary Fig. 3 is not right. The description for (a) is missing. Both the paragraph above the figure and the (c) caption mention “the radial binary mask (black frame)” or “dark field evaluation mask as black lines”, however, this mask is not shown in the (c) figure.
7. Page 4, "Figure" can be abbreviated as "Fig." except at the start of a sentence. Please change “Fig. 3 shows ...” to “Figure 3 shows ...” or “Supplementary Fig. 3 shows ...”.

Reply: We thank the Reviewer once again for these very helpful comments and have revised the manuscript accordingly. Points 1,2,4,5,7: corrections made as suggested, self explanatory. Point 3: this was indeed at typo, which has been corrected. Point 6: The mask for darkfield analysis is now shown as a black frame in Supplementary Fig. 3 b, the mask used for PCA is shown as red dashed circles in Supplementary Fig. 3 c. Moreover, the figure was rearranged a little bit in order to increase the reading flow in the supplementary methods section.

Requests/Reply Editorial Board:

There are a number of requirements that need to be addressed. We will be unable to proceed with acceptance of your manuscript until it adheres to these requirements. Please use the tracked changes feature of Microsoft Word to make these changes.

Please provide a point-by-point response to these points with your submission.

Reply: We were not able to use the tracked changes feature of Microsoft Word since the article is written in Latex/Latex code.

1. transparent peer review system

Reply: We agree with having the transparent peer review.

2. editorial policy checklist

Reply: We have uploaded the corresponding checklist together with our revised article.

3. policies and format requirements

Reply: We have now adapted the article and SI to these requirements.

4. data availability statements and data citations policy
5. data sources
6. hyperlink and DOI

Reply to Point 4-6: We have now uploaded the main datasets of figure 2 (main article) to an open access scientific repository, zenodo.org (supported by CERN). Uploaded data is saved and will be published under a reserved DOI as soon as this MS is accepted, We have incorporated the corresponding reference in the data availability statement of the main article.

7. third-party images

Reply: The manuscript contains no third-party images.

8. relevant details

Reply: All relevant details are given in the MS and the SI.

9. 15 words intitle
10. abstract to 150 words
11. article sections
12. secondary subheadings
13. subheadings in the Results and Methods fewer than 60 characters
14. remove Figures and move Figure Legends

Reply to point 9-14: We have adapted the article to all these issues accordingly following the even more detailed checklist of point 3.

15. full Methods section

Reply: The Methods section has been extended.

16. "data not shown"

Reply: We have removed this wording from the text.

17. sufficient information in Methods part

Reply: Sufficient information is given in the methods part and we have now circumvented the wording 'the procedures described in ref...' by adding clarifying details to the methods part.

18. speech marks

19. ambiguity in the mathematics

Reply to point 18-19: We have now adapted the used equations and units in text, figures and tables in this respect.

20. origins of cell lines

Reply: We have addressed this issue by adding more details to the methods part.

21. color scales

22. legend and title of figures

23. specific figure panels

24. subdivision of figure panels

25. abbreviations and symbols in figures

26. references format

27. 'Competing Interests' statement

28. style guidelines in supplementary information files

29. supplementary items

30. general citations to the Supplementary Information

31. Supplementary References

32. color-coded numbering

Reply to point 21-32: We have addressed all these formatting issues as best as we could as concerns the figures, text and references of the main article as well as text and sectioning of the supplementary information. A 'Competing Interests' statement has now been added, sections and captions of figures & tables have been adapted accordingly. We found, that requirements such as to include "a title that summarises the figure and does not refer to specific panels" of particular help, because it indeed upgrades the manuscript! As concerns the restriction of subdivisions of figure panels to 1a,b,c, we have adapted figure 1 of the main article and have removed our previous numbering in circles. While we still keep the color-code (red, green, blue box) in Fig.1(a), the different modalities are also indicated by headlines.

33. two-sentence editor's summary

- "X-ray techniques benefit from correlative imaging approaches, but combination with super-resolution microscopy has not been explored. Here the authors image the cardiomyocyte cytoskeleton by combining holographic x-ray imaging, x-ray scanning diffraction and STED in the same synchrotron end-station."

Reply: Please change the two-sentence editor's summary as follows:

"X-ray techniques benefit from correlative imaging approaches, but combination with super-resolution microscopy has not been explored. Here the authors image **the actin cytoskeleton of cardiac tissue cells** by combining holographic x-ray imaging, x-ray scanning diffraction and **STED microscopy** in the same synchrotron end-station."

SUBMISSION INFORMATION:

34. cover letter

35. point-by-point response to referees

36. LaTeX file

37. Production-quality of figures

38. figurestyle for colour-blind readers

39. final version of any Supplementary Information

40. 'Featured Image'

41. completed author checklist

42. signed copies of LTP

Reply to point 34-42: All required items are provided with the online submission.

Requests/Reply Editorial Board (email of Dr. Catherine Farrell sent on July 11th, 2018):

43. In your Supplementary Information, please change the heading 'Supplementary Notes' to 'Supplementary Note 1', and ensure that the citation in your article is also updated accordingly.
44. Please ensure that your supplementary figures and tables are cited in your article.
45. Please ensure that your abstract is given the heading 'Abstract'.

Reply to point 43-45: We have now adapted the main article and supplementary information to all these formatting issues.